# Generalizing Neural Additive Models via Statistical Multimodal Analysis

**Young Kyung Kim**                                            *yk206@duke.edu*
*Department of Electrical and Computer Engineering*
*Duke University*

**J. Matías Di Martino**                          *matias.di.martino@duke.edu*
*Department of Electrical and Computer Engineering*
*Duke University*

**Guillermo Sapiro**                            *guillermo.sapiro@duke.edu*
*Department of Electrical and Computer Engineering*
*Duke University*

**Reviewed on OpenReview:** *https://openreview.net/forum?id=xLg8ljlEba*

## Abstract

Interpretable models are gaining increasing attention in the machine learning community, and significant progress is being made to develop simple, interpretable, yet powerful deep learning approaches. Generalized Additive Models (GAM) and Neural Additive Models (NAM) are prime examples. Despite these methods' great potential and popularity in critical applications, e.g., medical applications, they fail to generalize to distributions with more than one mode (multimodal[1]). The main reason behind this limitation is that these "all-fit-one" models collapse multiple relationships by being forced to fit the data unimodally. We address this critical limitation by proposing interpretable multimodal network frameworks capable of learning a Mixture of Neural Additive Models (MNAM). The proposed MNAM learns relationships between input features and outputs in a multimodal fashion and assigns a probability to each mode. The proposed method shares similarities with Mixture Density Networks (MDN) while keeping the interpretability that characterizes GAM and NAM. We demonstrate how the proposed MNAM balances between rich representations and interpretability with numerous empirical observations and pedagogical studies. We present and discuss different training alternatives and provided extensive practical evaluation to assess the proposed framework. The code is available at https://github.com/youngkyungkim93/MNAM.

## 1 Introduction

Deep neural networks (DNN) achieve extraordinary results across several important applications such as object detection (Redmon et al., 2016; Girshick et al., 2014; Ren et al., 2015), object classification (He et al., 2016; Krizhevsky et al., 2017; Dosovitskiy et al., 2020), and natural language processing (Mikolov et al., 2013; Devlin et al., 2018; Brown et al., 2020). Yet DNN's popularity is still low in critical applications where miss-classification has high consequences or transparency is required for decision-making, e.g., to prevent unfairness toward certain groups; examples are medical-related risk estimation and machine learning (ML) based public policies. According to experts in these domains, one of the main factors limiting the adoption of DNN-based approaches is the lack of interpretability and trustworthiness associated with these algorithms (Shorten et al., 2021; Amarasinghe et al., 2020; Li et al., 2022). Even though several techniques have been

---

[1]In this paper, multimodal refers to the context of distributions, wherein a distribution possesses more than one mode.

proposed to increase the understanding of DNN (Agarwal et al., 2021; Ribeiro et al., 2016; Pedapati et al., 2020), medical professionals and policymakers still prefer simple models for which they can understand directly the factors that lead to a particular prediction. On the opposite end of DNN are algorithms such as linear regression and its multiple variants (Montgomery et al., 2021), which are simple and interpretable but lack the flexibility and high performance that DNNs have. Notably, linear models can not capture nonlinear relationships nor exploit numerous novel tools that efficiently optimize modern DNN approaches. A recent approach proposed by Agarwal et al., named Neural Additive Models (NAM), generalizes Additive Models (GAM) and achieves an interesting balance between interpretability and learning power. Individual features undergo nonlinear transformations independently, and these transformed features are merged in a regression-like paradigm, allowing the user to understand the weight of each factor leading to the prediction. This enables the algorithm to learn non-trivial relationships between the features and the target outcomes while leveraging powerful state-of-the-art optimization tools developed for deep learning.

Although most of the research on GAM has focused on minimizing the trade-offs between accuracy and interpretability (Nori et al., 2019; Zuur, 2012; Agarwal et al., 2021), addressing the lack of power for GAM and NAM in capturing multimodal relationships between input and target variables has been rare or nonexistent. This limitation is crucial especially when a dataset has multiple distinctive relationships between features and outputs. For example, imaging in the context of a medical application where we are predicting the glucose level $y$ using electronic health records (EHR) as input variables $x_1$, ..., $x_n$; let us assume there are two subpopulations identified by the variable $d \in \{0, 1\}$, which can be observed or a latent feature. For both cases, NAM would fail to capture a relationship in which $y$ is positively correlated with $(x_1 | d = 0)$ but is uncorrelated with $(x_1 | d = 1)$. This is due to NAM only learning one deterministic relationship between inputs and outputs. When $d$ is a latent variable, NAM will fail to differentiate them and collapse two relationships into one average deterministic relationship. Even if $d$ is an observed feature, NAM will fail to differentiate them as a DNN assigned for $X_1$ does not take $d$ as an input to have information on two subpopulations.

To address this while preserving the virtues of NAM, we propose a probabilistic Mixture of Neural Additive Models (MNAM). The main idea is to apply mixture density networks (MDN), a neural network with mixture of $k$ Gaussian distributions as an outcome, as a linking function for GAM to model the relationship between inputs and outcomes in a multimodal fashion and associating a probability to each mode. The probability of each mode enables the model to be flexible in representing multiple subpopulations as MNAM is able to activate accurate relationships for certain subpopulations by increasing their probability.

Figure 1 illustrates the power and flexibility of MNAM. These strengths are also illustrated in Section 3 through applying MNAM on real datasets. Such flexibility will be especially crucial in decision-making with high consequences. For example, for analyzing the side effects of medicine, 99% of participants might have steady glucose levels but 1% might have high and dangerous glucose levels after taking a medicine. NAM will collapse both levels into one indicating no side effects on average, but MNAM will accurately show, with probability, two glucose levels of different subpopulations.

It is important to highlight the interpretability of the model. Similar to NAM having a one fixed relationship between input features and output variables, MNAM will have fixed multiple relationships, which makes the model interpretable. Only the probability of each mode will change from the change in other features, which indicates changes in a subpopulation.

Our main contributions are: (i) we identify the overlooked limitation that GAM and NAM have when they are trained with a dataset that has multiple subpopulations; (ii) we provide a practical alternative to solve the critical problem of "one-fits-all" standard in interpretable DL approaches; (iii) we propose a model called MNAM that could learn multiple relationships among subpopulations for the solution; (iv) we propose a method to train MNAM, with objectives to learn multiple relationships and activate one or more matching relationships for a given subpopulation; and (v) we demonstrate MNAM is more expressive in accuracy and flexible in interpretability compared to NAM. We describe the proposed method in Section 2. Section 3 presents empirical evidence and pedagogical studies, showing strengths of MNAM. We discuss related work in Section 4 and limitations in Section 5. Finally, we provide conclusions in Section 6.

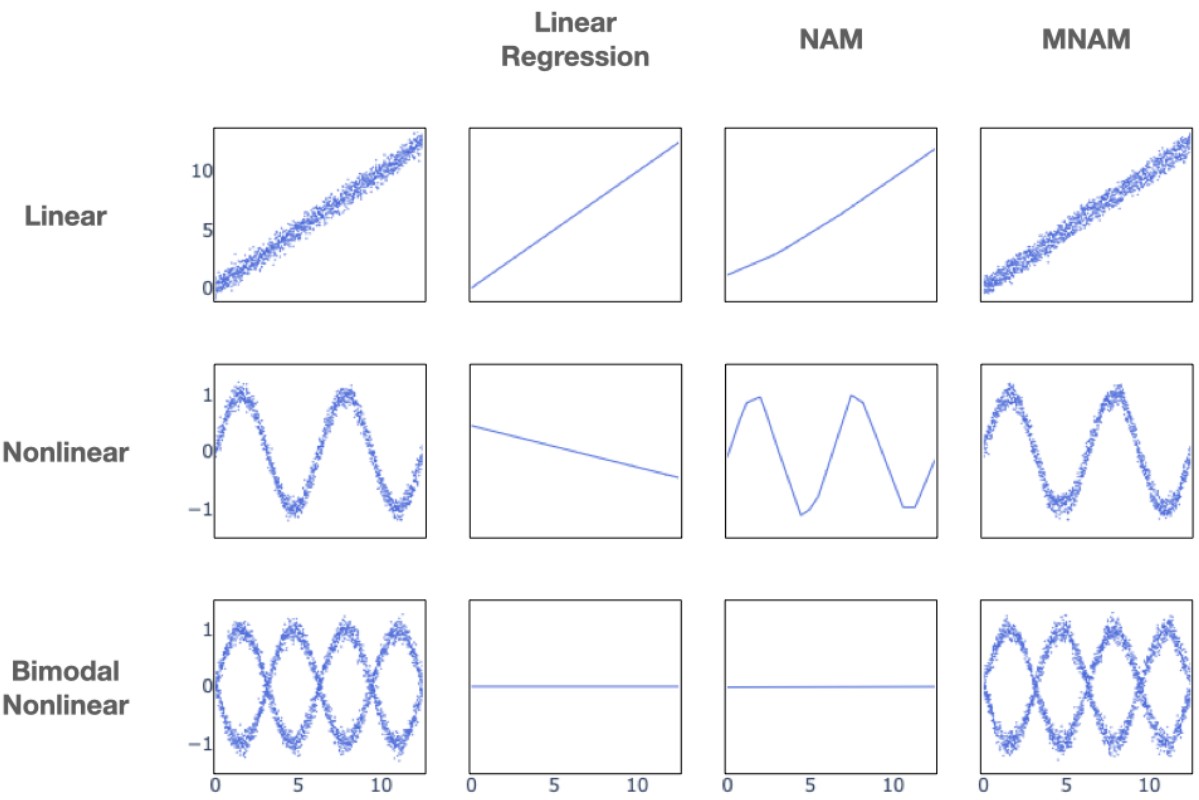

Figure 1: Linear regression, NAM, and MNAM on linear, nonlinear, and bimodal data. The left column illustrates the input for three datasets. The columns illustrate the representations learned by linear regression, NAM, and MNAM, respectively. As expected, linear regression fails to learn datasets with nonlinear relationships. NAM fails to learn datasets with relationships that have more than one modality, and only MNAM is able to learn nonlinear and multimodal relationships.

## 2 Method

### 2.1 Architecture

MNAM produces an outcome that is a mixture of $k$ Gaussian distributions. These distributions can be represented as $(\mathcal{N}_1(\mu_1, ,\sigma_1^2), ..., \mathcal{N}_k(\mu_k, ,\sigma_k^2), \pi_1, ..., \pi_k)$. Here, $\mathcal{N}_i(\mu_i, ,\sigma_i^2)$ represents a standard Gaussian distribution with mean $\mu_i$ and standard deviation $\sigma_i$, while $\pi_i$ denotes the associated probability. The purpose of utilizing a mixture of $k$ Gaussian distributions, a universal approximator for any density distribution, as the outcome for the model, is to represent multiple relationships between inputs and outcomes. For representation, one or more Gaussian distributions are assigned to each input-output relationship, and MNAM activates specific relationships for given subpopulations of the input by increasing the probability of the appropriate Gaussian distributions. This property is significant because it allows us to successfully capture and represent modes for relationships within various subpopulations in the dataset. Importantly, this approach does not require prior knowledge of the number of modes, as long as the model has a sufficiently large value of $k$ relative to the number of modalities. In contrast, models like GAM and NAM are unable to accurately represent multimodal relationships since they only provide a single estimated outcome per representation. In Section 3.3, we provide a demonstration of this property and also compare the representation of multimodal relationships between MNAM and NAM through a pedagogical example.

Similar to NAM, MNAM predictions are built from a linear combination of embeddings $Z_i$ of each input feature $X_i$ mapped through a neural network. In contrast with NAM, MNAM embedding consists of parameters for $k$ Gaussian distributions and a latent variable for predicting the probability of the mixture of $k$ Gaussian distribution models $(\mathcal{N}_{j,1}(\mu_{j,1}, \sigma^2_{j,1}), ..., \mathcal{N}_{j,k}(\mu_{j,k}, \sigma^2_{j,k}), Z^\pi_j)$. The left index $j$ is a reference to one of the input features and the right index of the Gaussian distributions is a reference to the number of components for the mixture. As shown in Equation 1, we compute the mean and variance of the Gaussian distributions for the MNAM outcome by linearly combining the mean and variance of matching components for Gaussian distributions of features' embedding.

$$\mathcal{N}_{1,i}(\mu_{1,i}, \sigma^2_{1,i}) + ... + \mathcal{N}_{n,i}(\mu_{n,i}, \sigma^2_{n,i})$$
$$= \mathcal{N}(\sum_{j=1}^n \mu_{j,i}, \sum_{j=1}^n \sigma^2_{j,i}) = \mathcal{N}_i(\mu_i, \sigma^2_i) \tag{1}$$

The linear property is crucial for the interpretability of additive models. In the case of GAM and NAM, where the output is a linear combination of non-linearly transformed features, the models are interpretable because one can observe how changes in a feature impact the outcome. Therefore, the mixture of Gaussians as the outcome has practical advantages. By leveraging the linear property of a mixture of Gaussians, MNAM can capture the magnitude of changes in the overall mean and uncertainty of predictions in response to variations in a feature.

Latent variables for predicting the probability of the mixture of $k$ Gaussian distributions for all features' embeddings will be the input for a separate neural network that predicts the mode associated with the output. This neural network will learn to identify which subpopulation is being represented based on input from all features, and activate the correct relationships by assigning a high probability to the matching Gaussian distributions. The description of how MNAM computes predictions is summarized in Algorithm 1 and the comparison of the architecture for NAM and MNAM is illustrated in Figure 2.

---

**Algorithm 1** Mixture Neural Additive Models

**Input:** Data: $(X_1, ...X_n)$, Number of Features: $n$, Number of Gaussian Distributions: $k$, Neural Networks for Feature Transformation: $(f_1, ..., f_n)$, Neural Network for Probability: $g$
**Output:** Mixture of Gaussian Distributions: $\mathcal{N}_1(\mu_1, \sigma^2_1), ..., \mathcal{N}_k(\mu_k, \sigma^2_k), \pi_1, ..., \pi_k$
**for** $i = 1$ **to** $n$ **do**
  $\mathcal{N}_{i,1}(\mu_{i,1}, \sigma^2_{i,1}), ..., \mathcal{N}_{i,k}(\mu_{i,k}, \sigma^2_{i,k}), Z^\pi_i = f_i(X_i)$
**end for**
**for** $i = 1$ **to** $k$ **do**
  $\mu_i = \sum_{j=1}^n \mu_{j,i}$
  $\sigma^2_i = \sum_{j=1}^n \sigma^2_{j,i}$
**end for**
$\pi_1, ..., \pi_k = g(Z^\pi_1, ..., Z^\pi_n)$

---

## 2.2 Training and Optimization

As mentioned in Section 1, state-of-the-art optimization tools for deep learning are applicable for training MNAM. For this work, we used Adam (Kingma & Ba, 2014) with a learning rate decreasing by 0.5% for each epoch. The objective of the training and optimization of MNAM is to assign one or more Gaussian distributions to each relationship in the dataset. Another objective is to learn to identify subpopulations from the given features to activate the correct relationship associated with the given sample. We devise hard-thresholding (HT) and soft-thresholding (ST) algorithms for the given objectives. The HT algorithm trains or updates a single mode or a Gaussian distribution with a minimum loss, while the ST algorithm updates all $k$ modes with weights computed by the likelihood of each mode on the label. Our optimization objective shares similarities with the work proposed in the context of Mixture Density Networks (MDN) (Bishop, 1994). Bishop discussed probabilistic formulations for deep neural networks that use a mixture of

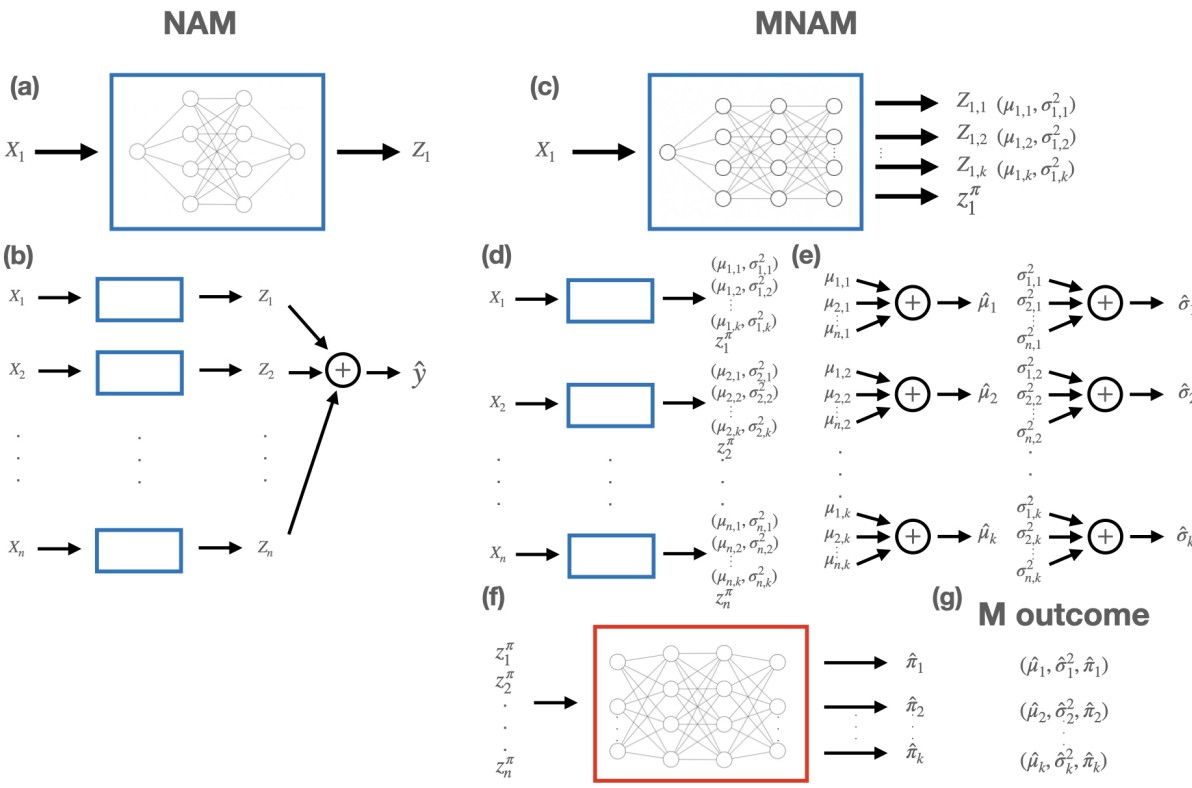

Figure 2: Illustrative schemes of NAM and MNAM network architectures. As shown in (a) and (b), NAM independently maps features into embedding through neural networks and then linearly combines embeddings for a prediction. Similar to NAM, MNAM independently maps features into embeddings through neural networks. The difference is that embedding consists of $k$ Gaussian distributions and a latent variable for predicting probabilities for a mixture of the $k$ Gaussian distributions, which is illustrated in (c) and (d). (e) illustrates linear combinations of each component of the Gaussian distributions for all features' embeddings. (f) depicts the mapping of latent variables for a mixture of $k$ Gaussian distributions $(Z_1^\pi, Z_2^\pi, ..., Z_n^\pi)$ into probabilities for the mixture of $k$ Gaussian distributions through a neural network. (g) is an example of the outcome for MNAM, which is the mixture of $k$ Gaussian distributions.

Gaussians for outcomes and provided alternative optimization methods. MDN is optimized by maximizing the log-likelihood of the data with respect to their mixture of Gaussian outputs,

$$\log(\sum_{j=1}^{n} \pi_i \times \mathcal{N}_i(Y|\mu_i, \sigma_i^2)). \tag{2}$$

In contrast to MDN, we optimize two terms, one that focuses on fitting one of the modes to a particular sample and the other that learns to predict the mode to which the instance belongs (this is described next in Section 3.2). Moreover, despite the similarities between our training objective and MDN log-likelihood, we observed that our proposed method is significantly more stable, easier to train in practice, and solves some of MDN's known and reported issues (Choi et al., 2018; Makansi et al., 2019). We expand on this in the coming sections.

Among the three algorithms discussed next, we chose the HT algorithm since it is more numerically stable and performs better in likelihood metrics; this is demonstrated in the empirical experiment in Section 3.2.

### 2.2.1 Hard-Thresholding (HT) Algorithm

In order to address the two training objectives, the algorithm calculates two separate losses. For the objective of assigning one or more Gaussian distributions to each relationship, the algorithm computes the Gaussian negative log-likelihood (GNLL) loss for each Gaussian distribution of the outcome against a label. Among $k$ losses, only the minimum factor will be used to compute the total loss, which means only weights used to compute the minimum loss are updated via the backpropagation. This enables the model to assign one or more Gaussian distributions to learn each relationship. For the second objective, which involves learning to identify subpopulations, the algorithm computes the cross-entropy loss between the probabilities of a mixture of Gaussian distributions for a prediction and the index number of the Gaussian distribution with the minimum loss. This loss measures how well MNAM activates the corresponding Gaussian distribution for the input and enables the model to learn to identify subpopulations for a given input to increase the probability of the correct Gaussian distribution for representation. Algorithm 2 summarizes the proposed training algorithm. It is important to highlight that the proposed learning method is unsupervised, in the sense that the data subgroups do not need to be known or defined in advance.

---

**Algorithm 2** Hard-Thresholding (HT) Algorithm

---

**Input:** Data: $(X, Y)$, MNAM: $f$, GNLL loss: $g$, Cross-entropy loss function: $h$, Rate for cross-entropy loss: $\lambda$

$\mathcal{N}_1(\mu_1, \sigma_1), ..., \mathcal{N}_k(\mu_k, \sigma_k), \pi_1, ..., \pi_k = f(X)$

$min\_loss = 0$

**for** $i = 1$ **to** $k$ **do**

    $gau\_loss = g(\mathcal{N}_i(\mu_i, \sigma_i), Y)$

    **if** $min\_loss > gau\_loss$ **then**

        $min\_loss = gau\_loss$

        $min\_index = i$

    **end if**

**end for**

$prob\_loss = h((\pi_1, ..., \pi_k), min\_index)$

$total\_loss = Min\_loss + \lambda \cdot prob\_loss$

---

### 2.3 Soft-Thresholding Algorithm

Similar to the EM algorithm (Dempster et al., 1977), the ST algorithm has expectation and maximization steps for training. In the expectation step, we compute the posterior probability of subpopulations $P(Z = k|X, Y)$. As shown in Equation 3, we compute the posterior probability as,

$$\begin{aligned} P(Z = k|X, Y) &= \frac{P(X, Y|Z = k)P(Z = k)}{P(X, Y)} \\ &= \frac{P(X, Y|Z = k)P(Z = k)}{\sum_{i=1}^{k} P(X, Y|Z = k)P(Z = k)}, \end{aligned} \tag{3}$$

where $P(X, Y|Z = k)$ is the likelihood of $k^{th}$ Gaussian distribution for the given input, and $P(Z = k)$ is the prior probability of a subpopulation predicted from MNAM. In the maximization step, we update the weights of MNAM to maximize posterior probability of the subpopulations. First, we compute GNLL losses for all Gaussian distributions, and then GNLL losses for all the Gaussian distributions are linearly combined with weights matching their posterior probabilities from the expectation step. This ensures weights used to compute Gaussian distribution with a higher likelihood are updated more. Cross-entropy loss between the prior probability predicted from MNAM and the posterior probability computed in the expectation step is computed with a similar purpose as in the HT algorithm. Algorithm 3 summarizes the proposed training algorithm.

---

**Algorithm 3** Soft-Thresholding (ST) Algorithm

---

**Input: Data $(X, Y)$, MNAM $f$, GNLL loss $g$, Crossentropy loss function $h$, Rate for cross-entropy loss $\lambda$**

$\mathcal{N}_1(\mu_1, \sigma_1), ..., \mathcal{N}_k(\mu_k, \sigma_k), \pi_1, ..., \pi_k = f(X)$

**for** $i = 1$ **to** $k$ **do**

    $gau\_loss_i = g(\mathcal{N}_i(\mu_i, \sigma_i), Y)$

    $gau\_like_i = p(Y; \mu_i, \sigma_i)$

**end for**

$mar\_prob = \sum_{j=1}^{k} gau\_like_j \cdot \pi_j$

$\hat{\pi}_1, ..., \hat{\pi}_k = \dfrac{gau\_like_1 \cdot \pi_1}{mar\_prob}, ..., \dfrac{gau\_like_k \cdot \pi_k}{mar\_prob}$

$gau\_loss = \sum_{i=1}^{k} gau\_loss_i \cdot \hat{\pi}_i$

$prob\_loss = h((\pi_1, ..., \pi_k), (\hat{\pi}_1, ..., \hat{\pi}_k))$

$total\_loss = gau\_loss + \lambda \cdot prob\_loss$

---

## 2.4 Regularization

Similar to NAM, all regularization methods for deep learning can be applied to MNAM, including weight decay, dropout, and output penalty. For this study, we utilized weight decay and output penalty.

## 3 Experiments

### 3.1 Empirical Observations

#### 3.1.1 Datasets

We evaluate six datasets: the California Housing (CA Housing) (Pace & Barry, 1997), the Fair Isaac Corporation (FICO) (FICO, 2018), the New York Citi Bike (BIKE) (Vanschoren et al., 2013), the Medical Information Mart for Intensive Care (MIMIC-III) (Johnson et al., 2016), the US Census data on Income (ACS Income) for California (Ding et al., 2021), and the US Census data on Travel time (ACS Travel) for California (Ding et al., 2021).

The CA Housing dataset has the task of predicting housing prices and it consists of eight features.

**FICO**: The FICO dataset has the task of predicting credit scores and it consists of 24 features.

**BIKE**: The BIKE dataset has the task of predicting the duration of trips and it consists of four features. Due to limited computation resources, we dropped data points that had more than 4000 seconds of duration for a bike trip and sampled 25% of the remaining dataset for analysis.

**MIMIC-III**: MIMIC-III dataset has the task of predicting the length of hospitalization for patients and it consists of various static and dynamic features. For comparing NAM and MNAM, we have used only static features, which consist of seven features.

**ACS Income**: The ACS Income dataset for California has the task of predicting income and it consists of ten features.

**ACS Travel**: The ACS Travel dataset for California has the task of predicting travel time to work and it consists of 16 features.

#### 3.1.2 Training and Evaluation

Similar to how the original paper trained NAM, we used Bayesian optimization (Močkus, 1975) to finetune variables to train NAM and MNAM. Learning rate, weight decay, and output penalty are finetuned for NAM. Learning rate, weight decay, output penalty, number of Gaussian distributions, and lambda for the cross-entropy loss are finetuned for MNAM. For both models, we utilized early stopping to reduce overfitting.

Optimized parameters from Bayesian optimization can be found in the table from Appendix A. We used a 5-fold cross-validation for CA Housing, FICO, and MIMIC-III datasets, and a 3-fold cross-validation for BIKE, ACS Income, and ACS Travel datasets. For evaluation, we trained 20 different models by randomly splitting the train set into train and validation sets for each fold. We ensembled 20 models to evaluate on the test set. We calculated confidence intervals for each model, based on the mean and variance from the $n$-fold cross-validation.

To compare between deterministic and probabilistic models, we use the mean absolute error (MAE). However, the MAE fails to account for the uncertainty in predictions made by probabilistic models. Even if a probabilistic model accurately predicts a true distribution for the label distribution, it may still receive the same MAE score as a deterministic model if it is correct in predicting the mean of the label distribution. To address this limitation, we transformed a deterministic model into a probabilistic one, fitting a global variance, which is determined by computing the mean of the variances between the actual labels and the predicted labels from the training set. We then utilized mean likelihood (LL) as a metric for comparision, to emphasize the importance of having multimodal compared to unimodal distribution as an outcome. To have an extensive comparison between unimodal and multimodal models, we also trained NAM with tuned variance, which we will refer to as probabilistic Neural Additive Models (pNAM). We trained pNAM by setting $k = 1$ within the MNAM framework. Theoretically, pNAM offers enhanced flexibility compared to standard NAM approaches utilizing a globally fitted variance, as it can effectively represent data exhibiting heterogeneity of variances.

### 3.1.3 Results

Table 1 displays the MAE and LL scores of NAM, pNAM, and MNAM on datasets described above. MNAM consistently exhibited similar or superior MAE scores compared to other models across all six datasets. Moreover, MNAM showcased a significantly improved performance in terms of likelihood scores when compared to other models for all datasets, except for the FICO dataset. Notably, the optimized number of Gaussian distributions for MNAM was 1, which means that pNAM and MNAM are identical models. This finding underscores MNAM's remarkable ability to effectively learn the output distribution, surpassing both NAM and pNAM in this aspect.

Differences in performance between MNAM and NAM models differ greatly by datasets. Specifically, the discrepancy in likelihood scores between NAM models and MNAM is much more pronounced for the CA Housing dataset compared to the ACS Income dataset. Several explanations could account for this observation. Firstly, the CA Housing dataset might exhibit more intricate interaction relationships among its features, rendering it more challenging for NAM to accurately capture the underlying patterns without any interaction term learning. Conversely, MNAM, with its enhanced capability to model complex interactions, would demonstrate an improved likelihood score on such datasets. Secondly, the CA Housing dataset might possess modes that differ more significantly from one another, making it harder to fit using a single Gaussian distribution for NAM. In this scenario, MNAM would enhance the likelihood score by accommodating the complexity of interaction relationships and the differences among modes within the data.

| | $\sigma$-NAM | | pNAM | | MNAM | |
|---|---|---|---|---|---|---|
| Dataset | MAE↓ | LL↑ | MAE↓ | LL↑ | MAE↓ | LL↑ |
| CA Housing | $0.48 \pm 9e^{-05}$ | $0.43 \pm 9e^{-06}$ | $0.48 \pm 3e^{-05}$ | $0.58 \pm 6e^{-04}$ | $0.46 \pm 4e^{-05}$ | $0.73 \pm 0.001$ |
| FICO | $2.7 \pm 0.002$ | $0.073 \pm 8e^{-05}$ | $2.7 \pm 0.002$ | $0.084 \pm 2e^{-06}$ | $2.7 \pm 0.002$ | $0.084 \pm 2e^{-06}$ |
| MIMIC | $1.5 \pm 0.0002$ | $0.15 \pm 1e^{-07}$ | $1.5 \pm 0.0001$ | $0.15 \pm 3e^{-06}$ | $1.5 \pm 0.0003$ | $0.25 \pm 6e^{-05}$ |
| BIKE | $3.4 \pm 0.0005$ | $0.064 \pm 2e^{-09}$ | $3.4 \pm 0.0008$ | $0.069 \pm 3e^{-08}$ | $3.4 \pm 0.0006$ | $0.092 \pm 1e^{-06}$ |
| ACS Income | $37.2 \pm 0.003$ | $0.0049 \pm 2e^{-10}$ | $35.7 \pm 0.07$ | $0.011 \pm 4e^{-07}$ | $35.7 \pm 0.02$ | $0.013 \pm 4e^{-07}$ |
| ACS Travel | $15.6 \pm 0.0004$ | $0.013 \pm 6e^{-10}$ | $15.5 \pm 0.0008$ | $0.017 \pm 2e^{-08}$ | $15.5 \pm 0.002$ | $0.036 \pm 2e^{-05}$ |

Table 1: MAE and LL scores for NAM, pNAM, and MNAM on CA Housing, FICO, MIMIC, BIKE, ACS Income, and ACS Travel datasets. We highlighted best and second-best performances for each metric.

### 3.1.4 Interpretability

In this section, we visualize the relationships between features and labels, and how different relationships are activated from changes in subpopulations; we illustrate this for the CA Housing dataset. This illustrates the strength of the interpretability of MNAM. Relationships plots for other datasets can be found in Appendix C. As illustrated in Figure 3, MNAM is able to learn and represent multiple relationships between features and labels, which NAM fails to do as it collapses those relationships into a mean. Therefore, MNAM is more flexible in explaining and representing multiple relationships between features and labels by activating one or multiple of them. Another example of such visualization for CA Housing dataset is presented in Appendix B, where each relationship is plotted in separate subplots to enhance readability.

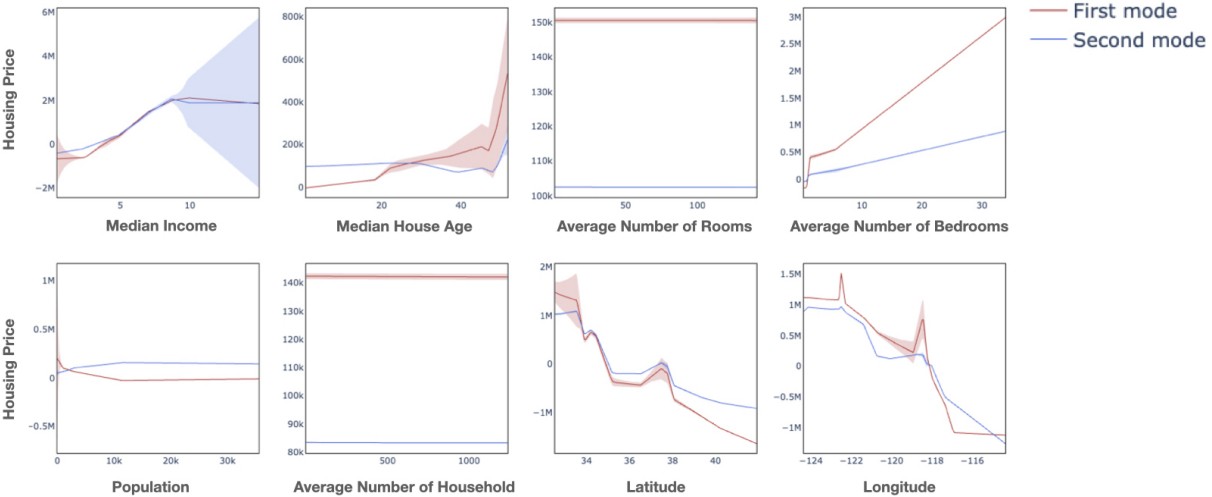

Figure 3: Two relationships between features and label learned by MNAM on CA Housing dataset. The solid lines represent the mean of these relationships, with highlighted regions indicating their uncertainties. By examining the solid lines and highlighted areas, users can analyze the changes in the relationship and the associated uncertainties between features and labels. Additionally, these subplots facilitate easy comparison, allowing users to discern whether the two relationships differ significantly.

Allowing multimodal data representations sheds light on non-trivial data relationships that are otherwise hidden in average "one-fit-all" models. For example, as illustrated in Figure 4, we identified that the price of a house could increase or decrease as the number of people in the neighborhood increases (the first column of Figure 4, illustrates the two modes recognized by MNAM). If we group the algorithm's output by median income (the first row of the second column represents the bottom one percent, and the first row of the third column represents the top one percent), we can recognize that one of the modalities is associated with higher income households and the other with lower income households. For example, the first row of the second column shows that the top mode is activated more frequently on this subgroup (darker blue represents higher frequency), suggesting that the larger the number the people in the neighborhood, the higher the house prices. The opposite can be recognized for the higher-income subgroups (see the first row of the third column). In other words, the output of the model suggests that for wealthier neighborhoods, the more people, the less expensive houses are, while the opposite occurs in poor communities. A similar story is illustrated when we group the algorithm's output by proximity to the beach (the second row of the second column represents inland, and the second row of the third column represents the area near the beach). The output of the model suggests that for areas near beaches, the more people the more expensive houses are, while the opposite occurs inland. Notice how these rich data interpretations would have been missed using NAM, where a "one-fit-all" model is optimized.

The strength of MNAM in identifying unfairness is more evident when trained on the MIMIC dataset. In Figure 5, the variance or the discrimination of the length of stay among different ethnicities significantly differ between the two modes recognized by MNAM (left graph of Figure 5). The first relationship, which is a red line, has more variance or discrimination among ethnicities compared to the second relationship, which is a blue line, in the length of stay. If we group the algorithm's output by admission type (the middle graph represents common admission and the right graph represents urgent admission), we recognize the model activates more on relationships with less discrimination among ethnicities with urgent admission and vice versa with common admission. In other words, there is more variance and discrimination in the length of stay among ethnicities for common admission compared to urgent admission. Overall, these findings highlight the capabilities of MNAM compared to NAM in identifying potential unfairness of a model as NAM will provide the same relationship for all subpopulations.

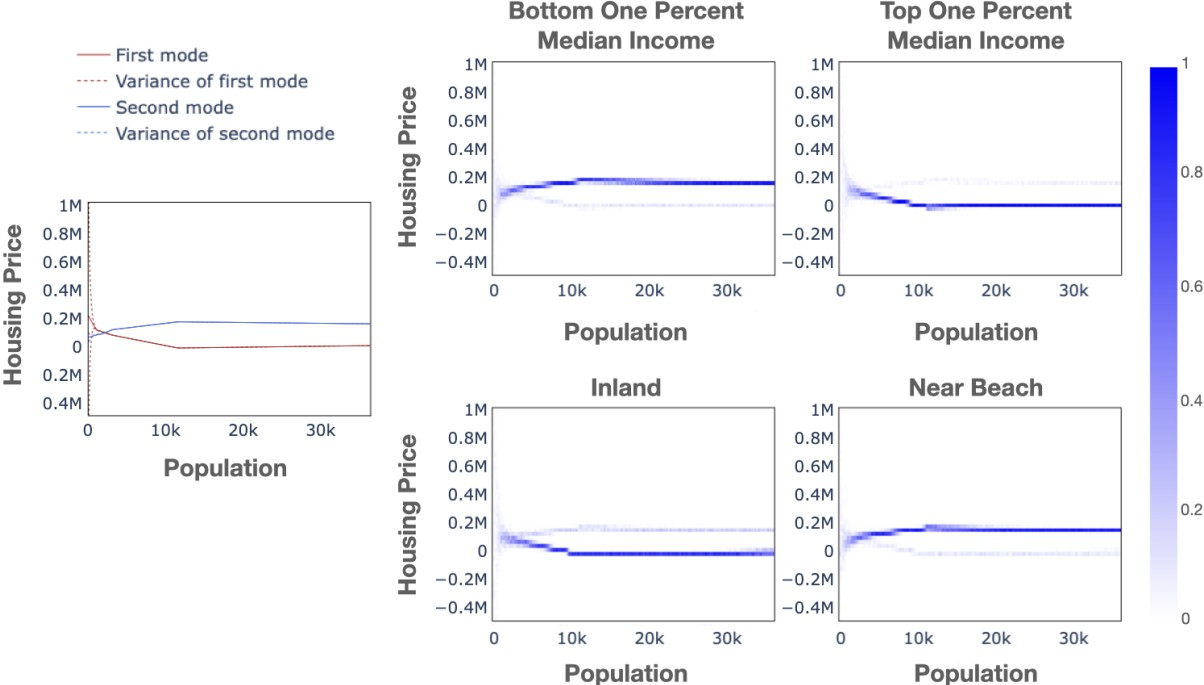

Figure 4: Line graph and heatmaps on the relationship between housing price and population for CA Housing. The first column, a line graph, represents two modes recognized by MNAM between housing prices and populations. Second and third columns, heatmaps, represents changes in the activation of two modes from changes in features of interest. The first row represents changes in median income and the second row represents changes in proximity to the beach. Except for each row's feature of interest, all other remaining features have been fixed to their mean values. The magnitude of the mode's activation is illustrated through the intensity of the color in the heatmaps. Darker blue represents higher activation of a mode. The blue color bar represents the magnitude of a mode's activation.

### 3.1.5  Training Efficiency

Table 2 shows the comparison of average training time, training time per epoch, and the number of epochs required to train NAM and MNAM on different datasets. As expected, MNAM takes longer to train per epoch than NAM because it has an additional neural network for computing mode weights. However, interestingly, MNAM was faster than NAM in training for half of the datasets, since MNAM required, in several, fewer epochs than NAM. Our hypothesis is that MNAM's assignment of modes to subpopulations effectively shrinks the space for the model to explore, resulting in fewer epochs needed for training, as

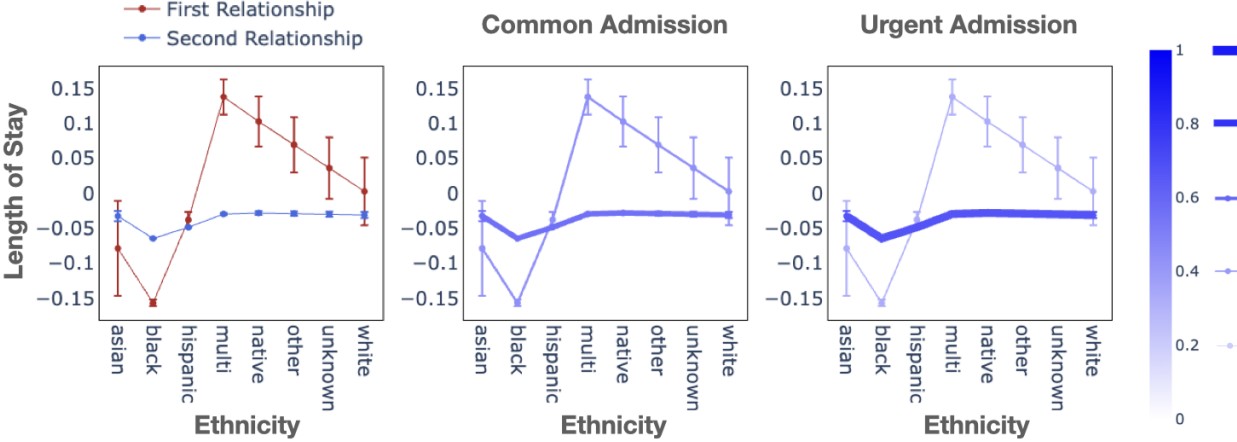

Figure 5: Line graphs of the relationship between the length of stay and the ethnicity for MIMIC with whisker representing a variance. The left graph represents two modes recognized by MNAM between the length of stay and the ethnicity. The middle and right graphs represent changes in the activation of two modes from changes in admission type. Except for admission type, all other remaining features have been fixed to mean values. The magnitude of the mode's activation is illustrated through the intensity of color and thickness of lines. The darker blue and thicker line represents higher activation of a mode. The blue color bar and different thickness of lines on the right side of the color bar represents the magnitude of a mode's activation.

each mode only needs to represent one subpopulation. In contrast, NAM has only one outcome that must represent multiple subpopulations, causing it to oscillate among subpopulations for representation during training. Overall, MNAM may be more efficient than NAM in terms of training time, despite having more parameters to compute.

| Dataset | Training Time (Seconds)↓ | | Training Time per Epoch (Seconds)↓ | | Number of Epochs↓ | |
|---|---|---|---|---|---|---|
| | NAM | MNAM | NAM | MNAM | NAM | MNAM |
| CA Housing | 636.38 | 411.73 | 0.72 | 0.84 | 885.36 | 489.85 |
| FICO | 326.26 | 337.54 | 0.78 | 0.85 | 417.34 | 397.67 |
| MIMIC | 173.10 | 278.58 | 0.93 | 1.09 | 186.64 | 256.53 |
| BIKE | 410.42 | 477.62 | 1.37 | 1.85 | 298.76 | 258.36 |
| ACS Income | 1460.31 | 896.41 | 3.22 | 4.47 | 453.65 | 200.40 |
| ACS Travel | 2411.51 | 946.55 | 3.86 | 5.75 | 624.43 | 164.58 |

Table 2: Comparisons of average training time in seconds, training time per epoch in seconds, and number of epochs between MNAM and NAM. We highlighted the best performance for each metric.

## 3.2 Comparison of Training Algorithms

In this section, we evaluated HT, ST, and a learning algorithm originally proposed by Bishop. This original algorithm optimizes the model to maximize the log-likelihood of a mixture of Gaussians distribution. We will refer to Bishop's algorithm as the original (ORG) algorithm. We assessed their performance based on numerical stability (NS), number of epochs per training (NE), and accuracy. Employing the dataset from the pedagogic study, we conducted 20 rounds of training using various learning rates to evaluate these metrics. NS was determined by calculating the percentage of successful training sessions without encountering gradient divergence. Accuracy was measured through MAE and LL computed on the test set. The evaluation results are presented in Table 3.

The HT algorithm exhibited superior performance in terms of NS and LL. One possible explanation for its better NS performance is that the HT algorithm exclusively considers the minimum GNLL loss term, whereas the ST and ORG algorithms incorporate all GNLL losses with associated weights for updates. These latter algorithms pass more gradients compared to the HT algorithm, introducing numerical instability during training, particularly in the initial stages when the loss is higher due to random weight initialization. Thus, the HT algorithm demonstrated successful training across all learning rates, while the ST and ORG algorithms struggled to converge at higher learning rates. This numerical instability in the ORG algorithm has been documented across various studies (Choi et al., 2018; Makansi et al., 2019). Furthermore, researchers have noted instances of mode collapse, which is also reflected in Table 3. While the ORG algorithm shows competence in predicting means, it sometimes struggles to effectively represent the multimodal distribution, potentially due to modes collapsing around the mean. In contrast, the HT algorithm attains the highest LL score, as it is able to accurately assign modes. Given that MNAM operates as a probabilistic model and the HT algorithm offers superior numerical stability, we chose the HT algorithm over the others.

| | HT ALGORITHM | | | | ST ALGORITHM | | | | ORG ALGORITHM | | | |
| LR | NS↑ | NE↓ | MAE↓ | LL↑ | NS↑ | NE↓ | MAE↓ | LL↑ | NS↑ | NE↓ | MAE↓ | LL↑ |
|---|---|---|---|---|---|---|---|---|---|---|---|---|
| 0.1 | 100% | 191.1 | 681.26 | 0.024 | 0% | - | - | - | 0% | - | - | - |
| 0.05 | 100% | 226.4 | 10.38 | 0.045 | 0% | - | - | - | 0% | - | - | - |
| 0.01 | 100% | 343.0 | 3.58 | 0.66 | 0% | - | - | - | 35% | 230.8 | 3.02 | 0.31 |
| 0.005 | 100% | 357.6 | 3.14 | 0.76 | 5% | 145.0 | 3.74 | 0.015 | 95% | 248.6 | 3.11 | 0.30 |
| 0.001 | 100% | 410.6 | 3.18 | 0.87 | 100% | 245.3 | 2.95 | 0.50 | 100% | 243.7 | 3.08 | 0.36 |
| 0.0005 | 100% | 503.4 | 3.19 | 0.89 | 100% | 362.3 | 2.94 | 0.66 | 100% | 247.7 | 3.07 | 0.36 |
| 0.0001 | 100% | 418.3 | 3.12 | 0.76 | 100% | 389.8 | 3.25 | 0.74 | 100% | 252.9 | 3.10 | 0.35 |
| $5e^{-05}$ | 100% | 394.4 | 3.34 | 0.61 | 100% | 342.8 | 3.36 | 0.57 | 100% | 269.0 | 3.17 | 0.29 |

Table 3: Comparision of Hard-Thresholding, Soft-Thresholding, Original algorithm on data from pedagogic study

### 3.3 Pedagogical Example

For pedagogical value and to further illustrate the differences between the original NAM and the proposed MNAM, we created a synthetic dataset with different subpopulations, which are differentiated by either observed or latent variables. NAM has limitations in accurately representing such dataset as it collapses four relationships between $X_1$ and $Y$ into one deterministic relationship by averaging them. When $X_2$ is an observed variable, NAM is not able to differentiate relationships, since a neural network assigned to $X_1$ does not take $X_2$ as input. The neural network for $X_1$ simply uses the average relationship for representation, which is shown when $X_2 = 0$ and $X_2 = 1$. The representation is worsened for NAM when variables that differentiate subpopulations are latent variables, which is the case for $X_2 = 2$ and $X_2 = 3$ in the synthetic dataset. NAM tries to represent multiple relationships with one relationship, as shown in the second column of Figure 6. MNAM overcomes such limitations as it is able to learn four relationships and activate the right relationships for each subpopulation. Another strength of MNAM is that as long as $k$ is larger than the number of relationships in a dataset, MNAM will be able to represent the relationships accurately. In other words, tuning $k$ is not critical, as long as its value is higher than the expected number of modes. Furthermore, MNAM is able to learn the uncertainty of each relationship, which NAM is unable to do. Described limitations of NAM and strengths of MNAM are illustrated in Figure 6.

Another advantage of MNAM is its ability to help discern whether a subpopulation is differentiated by observed or latent variables. This differentiation is possible by examining MNAM's output probabilities for each Gaussian distribution. If only one of the Gaussian distributions is activated, it suggests that the subpopulation differentiation is driven by observed variables. Conversely, if multiple distributions are activated, this implies the influence of a latent variable. For instance, in Figure 6, we observe that the subpopulation is differentiated by observed variables when $X_2 = 0$ and $X_2 = 1$. However, in scenarios where $X_2 = 2$ and $X_2 = 3$, the differentiation appears to be due to a latent variable. By analyzing MNAM's plots, users can determine the nature of subpopulation differentiation—observed or latent—based on the

count of activated Gaussian distributions. This insight is crucial for interpreting the model, as it provides an understanding of whether there is uncertainty in subpopulation identification and whether the collection of additional unobserved variables is necessary for more accurate characterization of these subpopulations.

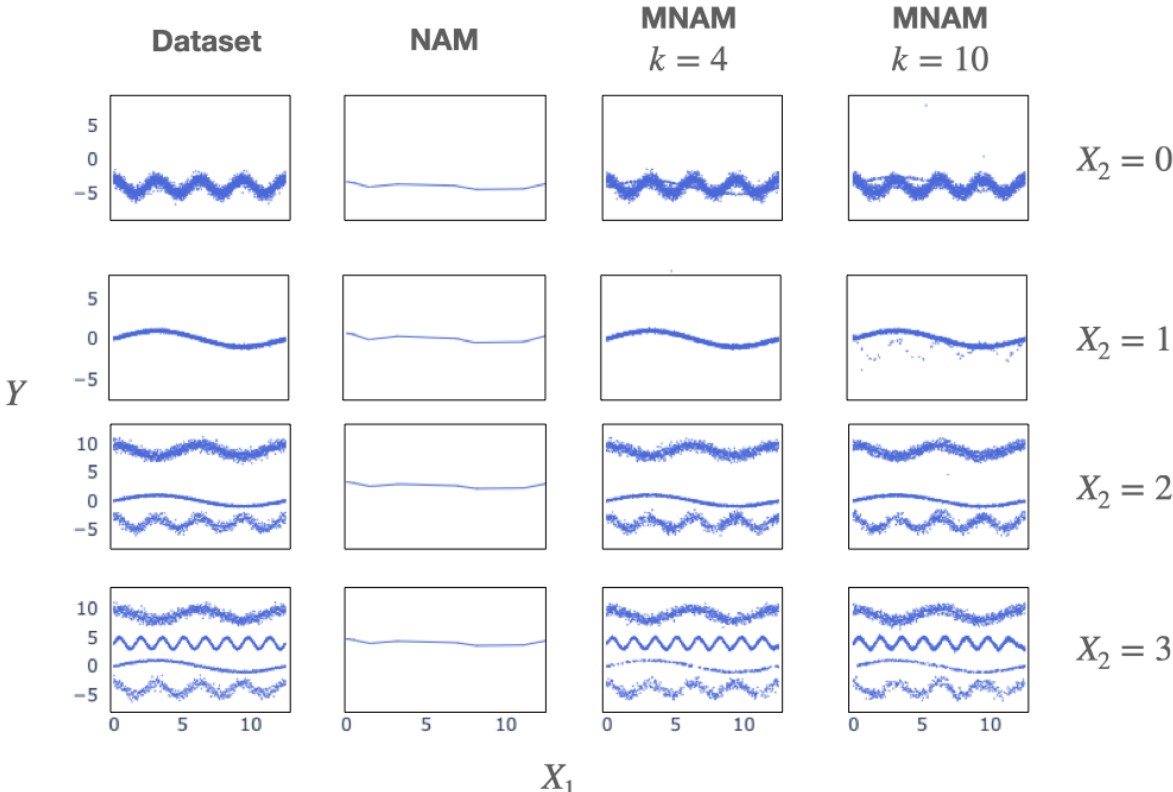

Figure 6: NAM versus MNAM on a dataset that has variables that identify subpopulations as observed and latent variables. The left column is a scatter plot for a dataset with different values of $X_2$. The remaining columns represent predictions from training on the dataset for NAM, MNAM with $k = 4$, and MNAM with $k = 10$. NAM clearly fails to represent the dataset as it collapses multiple relationships into one relationship. On contrary, MNAM with $k = 4$ and $k = 10$ accurately represents the dataset as it learns four relationships and activates the right ones for different values of $X_2$.

### 3.4 Trade-offs between Accuracy and Interpretability

We evaluated Linear Regression (LR); NAM; MNAM; Explainable Boosting Machine (EBM) (Nori et al., 2019), which is a form of Generalized Additive Models (GAM) with pairwise interaction terms; Gradient Boosting Trees (GBT) (Friedman, 2001; Pedregosa et al., 2011); and Mixture Density Networks (MDN) (Bishop, 1994). We used grid search for LR, EBM, GBT, MDN to finetune hyperparameters for training. Table 4 shows the MAE and LL scores for these five models. The order of the columns, left to right, represents an increase in complexity and a decrease in interpretability (here defined as a simple one-to-one relationship between input and output). The table is split into two, which are models with direct relationships and complex relationships (left and right respectively). LR, NAM, and MNAM are models with direct relationships because their feature and output relationships are fixed even from changes in other features. Meanwhile, EBM, GBT, and MDN are considered as models with complex relationships as their feature and output relationships changes from a change in other features due to their interaction terms.

The performance generally improves with an increase in the complexity of models, except for MDN. As mentioned above, numerical instability and mode collapse have been observed in various studies while training MDN (Choi et al., 2018; Makansi et al., 2019).

Excluding MDN, the MAE score improves from an increase in the complexity of models for most datasets (as expected). Interestingly differences in performances among models fluctuate greatly by datasets. This could be a result of datasets having different complexity. For example, models have similar performances on the MIMIC datasets. This could be due to the datasets being too simple to not even require nonlinearity or interaction terms of models for representations. In contrast, for the ACS Income dataset, the performance increases with an increase in complexity. This could be due to the dataset being more complex and requiring nonlinearity and more interaction terms with higher degrees for models to represent the dataset well. In contrast, the LL score did not exhibit improvement with an increase in complexity across all datasets, excluding MDN. For example, in the BIKE and MIMIC datasets, MNAM outperforms other more complex models. This potentially arises from the inherent need of the data to obtain a model with a multimodal output to accurately represent its underlying distribution.

| | Direct Input and Output Relationships | | | | | | | | Complex Input and Output Relationships | | | | | |
|---|---|---|---|---|---|---|---|---|---|---|---|---|---|---|
| Datasets | LR | | σ-NAM | | pNAM | | MNAM | | EBM | | GBT | | MDN | |
| | MAE↓ | LL↑ | MAE↓ | LL↑ | MAE↓ | LL↑ | MAE↓ | LL↑ | MAE↓ | LL↑ | MAE↓ | LL↑ | MAE↓ | LL↑ |
| CA Housing | 0.54 | 0.41 | 0.48 | 0.43 | 0.48 | 0.58 | 0.46 | 0.73 | 0.34 | 0.69 | 0.31 | 0.78 | 1.15 | 0.27 |
| FICO | 3.4 | 0.066 | 2.7 | 0.073 | 2.7 | 0.084 | 2.7 | 0.084 | 2.5 | 0.095 | 2.4 | 0.11 | 8.9 | 0.034 |
| MIMIC | 1.5 | 0.15 | 1.5 | 0.15 | 1.5 | 0.15 | 1.5 | 0.25 | 1.5 | 0.15 | 1.5 | 0.16 | 1.5 | 0.20 |
| BIKE | 3.65 | 0.061 | 3.4 | 0.064 | 3.4 | 0.069 | 3.4 | 0.092 | 3.4 | 0.065 | 3.4 | 0.065 | 3.93 | 0.083 |
| ACS Income | 40.0 | 0.0046 | 37.2 | 0.0049 | 35.7 | 0.011 | 35.7 | 0.013 | 33.3 | 0.0052 | 31.8 | 0.018 | 55.2 | 0.0081 |
| ACS Travel | 16.8 | 0.013 | 15.6 | 0.013 | 15.5 | 0.017 | 15.5 | 0.036 | 14.2 | 0.015 | 13.8 | 0.090 | 18.4 | 0.018 |

Complexity →

Interpretability ←

Table 4: MAE and LL scores for LR, NAM, MNAM, EBM, and GBT on CA Housing, FICO, MIMIC, BIKE, ACS Income, and ACS Travel datasets. The complexity of models increases from left to right and the interpretability of models increases from right to left.

# 4 Related Works

For interpretable models, GAM (Hastie, 2017) has been widely used. GAM transforms each feature by a function and linearly combines the transformed features, which enables features to have a fixed relationship with the output. For transforming each feature, various functions have been used such as boosted decision trees (Nori et al., 2019) and piecewise linear functions (Zuur, 2012). NAM (Agarwal et al., 2021) uses neural networks while GAM uses boosted decision trees (Lou et al., 2012; Guisan et al., 2002) to transform the features. Compared to those models, MNAM has multiple outputs with probability, instead of one single estimate. These multiple outputs enable the model to represent multiple subpopulations in the dataset. Furthermore, it is more flexible for interpretation as it is able to show multiple relationships between features and labels, and how different relationships are activated by changes in a subpopulation.

To overcome the limitations of GAM in representing multiple subpopulations, the Generalized Additive Model with Pairwise Interactions (GA2M) (Lou et al., 2013) was introduced, enhancing GAM by integrating interaction terms. The FAST algorithm was developed to efficiently rank all potential pairwise interactions, selecting only the top-k terms. This concept has inspired the development of various algorithms for ranking pairwise interactions. Researchers have applied sparsity and heredity constraints to identify significant features and interaction terms (Yang et al., 2021; Greenwell et al., 2023). The Constrained Generalized Additive 2 Model (CGA2M+) (Watanabe et al., 2021) further extends GA2M by incorporating higher-order interaction terms to improve expressiveness. However, since higher-order terms are not interpretable, they assess the validity of the interpretability of a model based on the importance score of the higher-order

interaction term. Despite ongoing research, the limitation of GA2M is that relationships between features and labels are not fixed due to its interaction terms, making the model less interpretable. The model requires users to read two graphs for interpretation. One is for a line graph on the relationship between label and features and another on the effect of the interaction terms.

Research on making ensemble methods interpretable has been progressing. In the realm of tree-based ensemble models, RuleFit (Friedman & Popescu, 2008) improves interpretability by extracting splitting rules from each node and assigning weights to these rules in a linear model. These rules also incorporate interaction terms. The Compressed Rule Ensemble (Nalenz & Augustin, 2022) approach further enhances interpretability by clustering similar rules and combining them into one. Another approach involves applying Bayesian model selection techniques to simplify tree ensembles, effectively reducing the number of decisions without losing the accuracy of the original model (Hara & Hayashi, 2018). The Interpretable Mixture of Experts (IME) (Ismail et al., 2023) model, a variant of the Mixture of Experts, extends these principles beyond tree-based ensembles. This model consists of multiple expert models and an assignment model, which assigns a specific expert to a given input. IME explores the trade-off between accuracy and interpretability by making the assignment model either interpretable or not while maintaining the interpretability of the expert models. Additionally, a hierarchical model is proposed that combines a deep neural network (not interpretable) with IME, thus providing interpretability to a subset of the dataset. Similar to one variant of IME, MNAM features multiple interpretable relationships and a non-interpretable assignment model, which is a neural network that predicts the weights of each relationship. The key difference is that IME uses one expert model for prediction instead of assigning weights to all expert models.

Furthermore, MNAM can show the relationship for minority subpopulations in the case when the subpopulation is differentiated by latent variables by assigning one of the $k$ relationships to the subpopulation. For deterministic models like NAM and DNN, they will fail as they are limited to showing only one relationship. This is demonstrated in Section 3.3 as the synthetic dataset for pedagogical experiment contains multiple subpopulations differentiated by latent variables. This can be critical, for example, in medical applications, where a drug might be effective in a certain subgroup of the population, tools like MNAM, would allow identifying from data modes or outliers that might not fit the general expected therapeutic trend.

Our work shares similarities with the Mixture Density Networks (MDN) proposed by Bishop in 1994. Bishop describes practical alternatives to use a mixture of $k$ Gaussian distributions to fit the outcome of a target variable using a neural network. In this work, the network loss is designed to optimize the posterior log-likelihood of the data. A notable difference between this work and the previously described work is that all the input features are processed to predict the target variable; hence, the method lacks the property GAM and NAM have to be able to map the output prediction to independent factors on the input features. Yet compared to NAM and GAM, MDN can handle multimodal distributions and does not collapse multimodal relationships into a single "all-fit-one" rule. Our work can be interpreted as combining those two paradigms, where we aim to preserve the good properties of the described methodologies. Integrating these two methods could prove beneficial for other machine learning models that rely on a single-estimate approach. In this paper, however, our focus is solely on NAM and GAM, as they are among the most widely recognized models for interpretability. Applying the tools here developed to other models is the subject of future research.

In our proposed method, we linearly combine features that have been separately transformed with neural networks to maintain independent relationships between features and output while fitting multimodal distribution with probabilistic interpretations. This approach allows the model to effectively learn from subpopulations. In contrast with MDN, we not only model the output distribution as multimodal but also assume an underlying subpopulation structure in the data and aim to detect from the input variables to which modality a given sample belongs. In addition, despite the similarities between our objective and MDN, extensive experimental evaluation suggests that our training objective is significantly easier to train, leading to faster, more stable, and better overall performance.

## 5   Limitations

MNAM trade-offs between accuracy and interpretability of a model. Increasing the number of $k$ Gaussian distributions for MNAM will increase accuracy. Yet, if the number of $k$ Gaussian distributions is large, then

it will be harder to interpret since there are too many possible relationships between features and outputs. The larger the number of $k$ Gaussian distributions in MNAM, the more the model will become similar to neural networks as it covers all separate relationships for all possible combinations of features. For our future work, we plan to explore methods such as the elbow method or other clustering techniques to find an optimal number for $k$ to balance between accuracy and interpretability.

As the MNAM model becomes more complex with an increased number of $k$ Gaussian distributions, a new challenge emerges: the model starts to treat outliers as an additional mode. This can be advantageous in certain domains. For instance, in medicine, recognizing a small subpopulation with significant side effects as a distinct mode could be pivotal for pinpointing critical issues. However, this approach may pose problems when outliers result from noise. In future work, we aim to develop more objective methods for outlier identification. A key strength of our model is that an outlier impacts only the Gaussian distributions assigned to it. Therefore, if we can accurately identify an outlier and its corresponding Gaussian in MNAM, we can exclude that Gaussian during the evaluation phase to mitigate the issue.

One limitation of the MNAM lies in its neural network, which predicts probabilities for a mixture of $k$ Gaussian distributions. This network's lack of interpretability makes it challenging to understand how changes in features affect the probabilities of the mixture. We can partially interpret the network by rigorously examining the model through various examples. Additionally, by modifying specific features, we can observe if there are variables that differentiate subpopulations, either observed or unobserved. Another approach involves examining the impact of each feature on the activation of probabilities for the mixture of $k$ Gaussian distributions. This can be done by dividing the data into bins of equal frequency and analyzing the average weight of each component across these bins. Examples of these examinations are provided in Section 3.3 and Appendix D. However, there are still limitations in fully interpreting the network. For future work, we aim to explore more interpretable models for predicting probabilities in mixtures of $k$ Gaussians, which would eliminate the need for such detailed examination.

MNAM learns means and variances of the outcome by minimizing the GNLL. However, this method is known to have limitations, as it may fail to effectively capture optimal means and variances of the outcome (Stirn et al., 2023). One observed issue is that the model exhibits a bias towards learning in regions with low variance, since the variance serves as a denominator for the gradient on the mean. Consequently, in regions with high variance, the gradient becomes small, impeding updates to the weights used to compute the mean. To address these limitations, researchers have introduced a technique where the variance is multiplied by the gradient (Stirn et al., 2023). This approach aims to prevent the high variance from causing the gradient to diminish to zero. In future works, we intend to apply this technique and explore other methods to further enhance the performance of the proposed model.

## 6    Conclusion

In this work, we introduced the Mixture Neural Additive Model (MNAM), an interpretable model more flexible than GAM and NAM, capable of capturing multimodal relationships between the input and output variables and with predictions that can be interpreted probabilistically. We showed that our ideas also connect with the Mixture Density Networks (MDN) framework. At the same time, MNAM preserves the desirable properties of NAM, and empirical evaluation suggests it trains more robustly and efficiently than MDN. We showed that our proposed model can be applied in practice and evaluated it across several real-world datasets and applications. Despite the recent progress and efforts in developing interpretable and robust deep learning alternatives, it is clear that there are still opportunities to improve the collection of tools that are available for practitioners working in critical applications such as health care and data-driven policymaking. Moreover, and despite great work like the one presented by Bishop in 1994, gaps remain between the theory and practice of deep learning, and the stability and practical application and training of neural networks remain a very active research topic (Choi et al., 2018; Makansi et al., 2019). Our work provides useful, robust, yet interpretable deep learning alternatives to help design and deploy data-driven solutions in critical applications. The code and pipelines to reproduce our work are open and can be found here: https://github.com/youngkyungkim93/MNAM.

## Acknowledgments

Work partially supported by ONR, NGA, NSF, and Simons Foundation. GS is also affiliated with Apple.

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

## A    Table of optimized parameters for MNAM

| Dataset | Learning rate | Weight decay | Output penalty | Number of Gaussian distribution | Cross-entropy loss |
|---|---|---|---|---|---|
| CA Housing | 0.009896 | 3.8512e-05 | 0.03363 | 2 | 0.6118 |
| FICO | 0.05 | 1e-06 | 0.0166 | 1 | 0.7762 |
| MIMIC | 0.01805 | 7.0946e-05 | 0.01908 | 2 | 0.7214 |
| BIKE | 0.01172 | 9.1022e-05 | 0.09256 | 6 | 0.3537 |
| ACS Income | 0.02873 | 9.13e-05 | 0.00167 | 4 | 0.494 |
| ACS Travel | 0.01894 | 9.3377e-05 | 0.0028 | 4 | 0.3634 |

Table 5: Optimized parameters for MNAM on six datasets

## B    Different Relationships plots on Housing

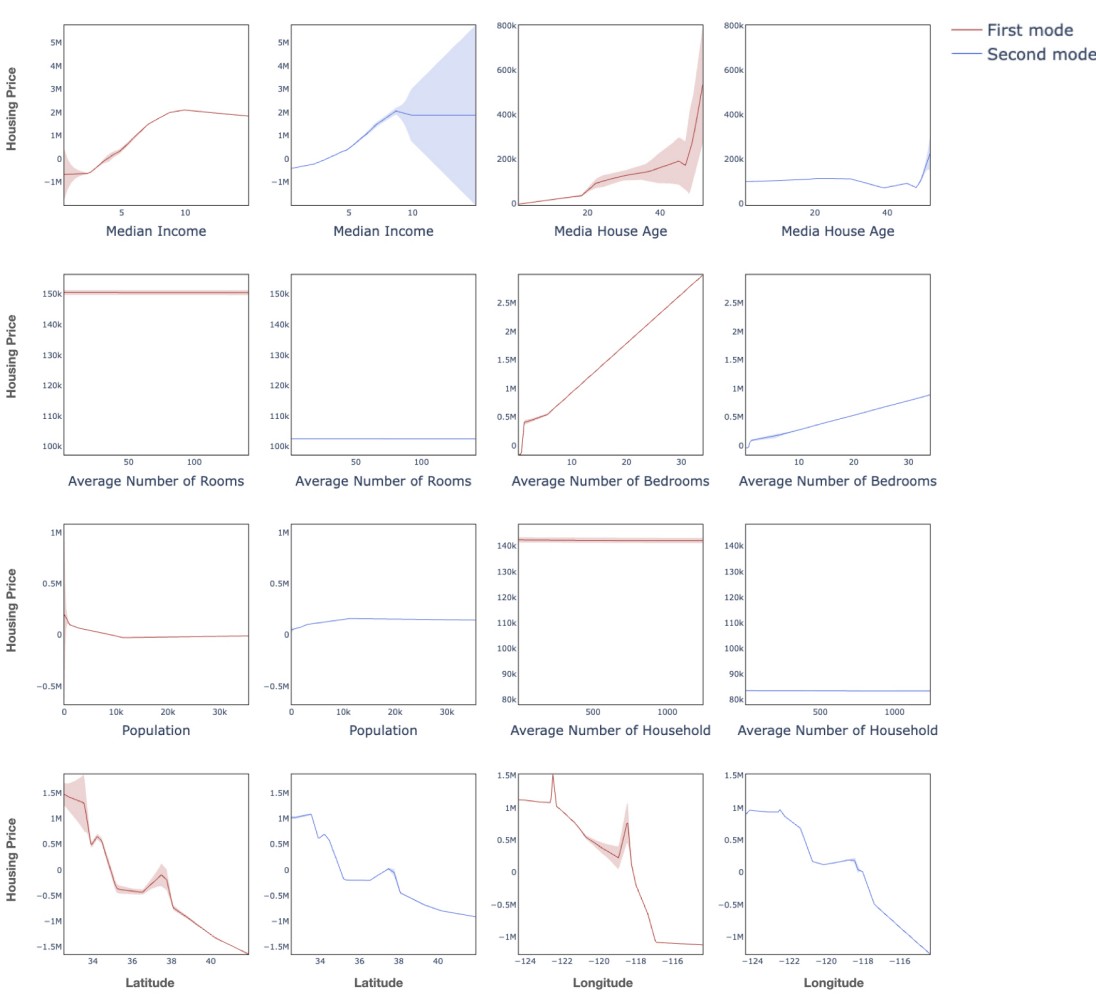

Figure 7: Learned relationships between features and labels for the MNAM on Housing datasets. Solid lines represent the mean of the relationships, while the highlighted regions represent their uncertainties. For enhanced readability, each subplot illustrates one distinct relationship.

## C Relationships plots on other datasets

### C.1 FICO

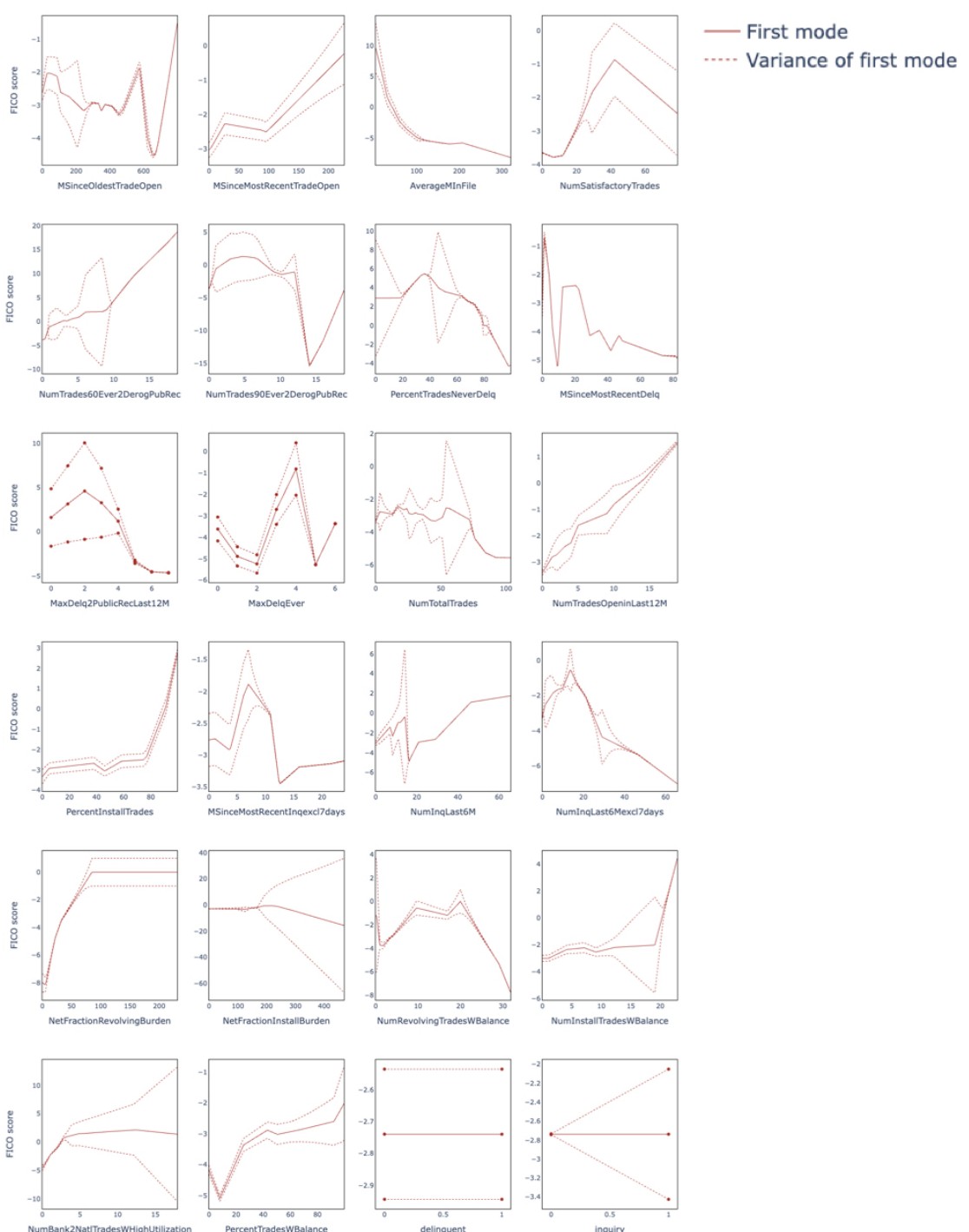

Figure 8: Learned relationships between features and labels for the MNAM on FICO datasets. Solid lines represent the mean of the relationships and dotted lines represent their uncertainties.

## C.2 MIMIC

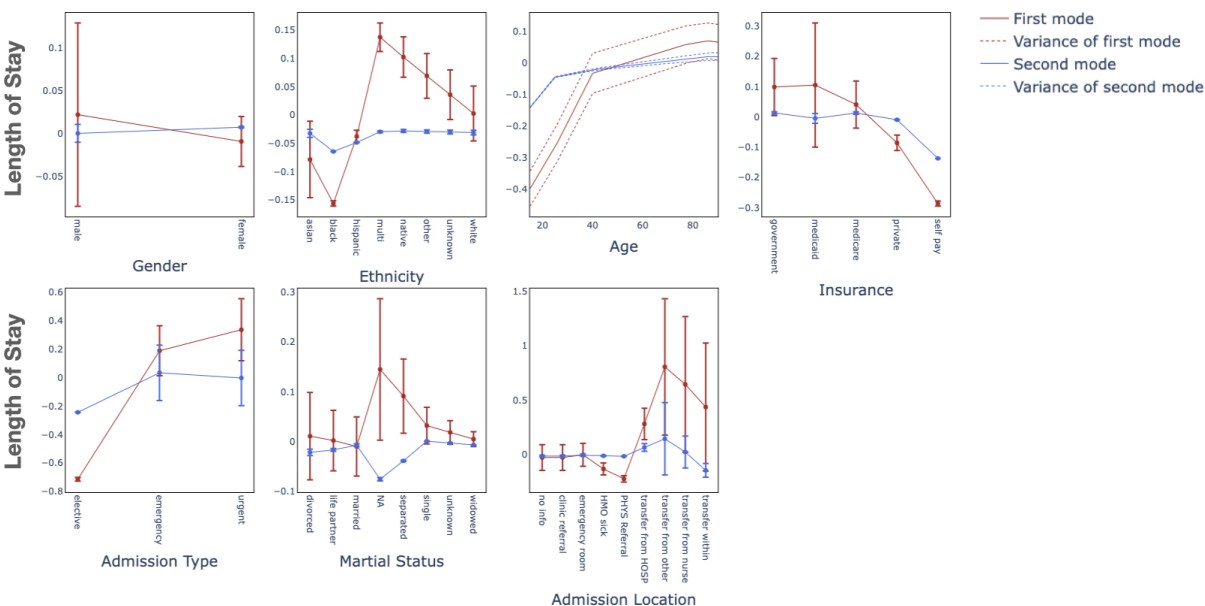

Figure 9: Learned relationships between features and labels for the MNAM on MIMIC datasets. Solid lines represent the mean of the relationships and dotted lines represent their uncertainties.

## C.3 BIKE

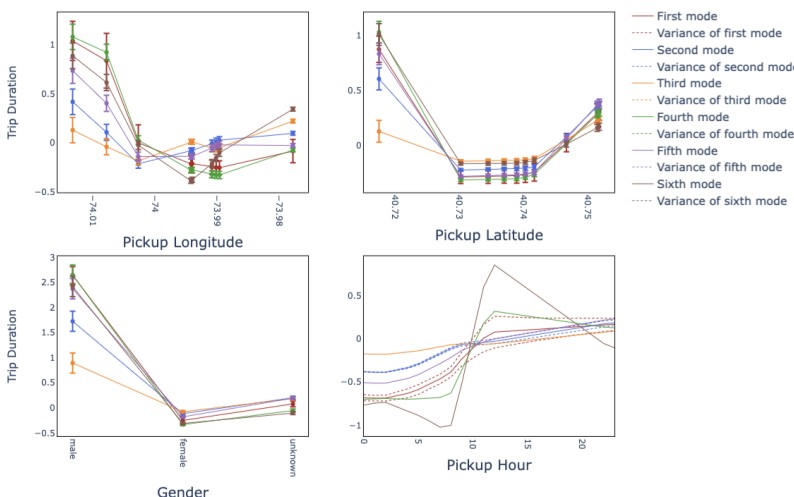

Figure 10: Learned relationships between features and labels for the MNAM on BIKE datasets. Solid lines represent the mean of the relationships and dotted lines represent their uncertainties.

## C.4 ACS Income

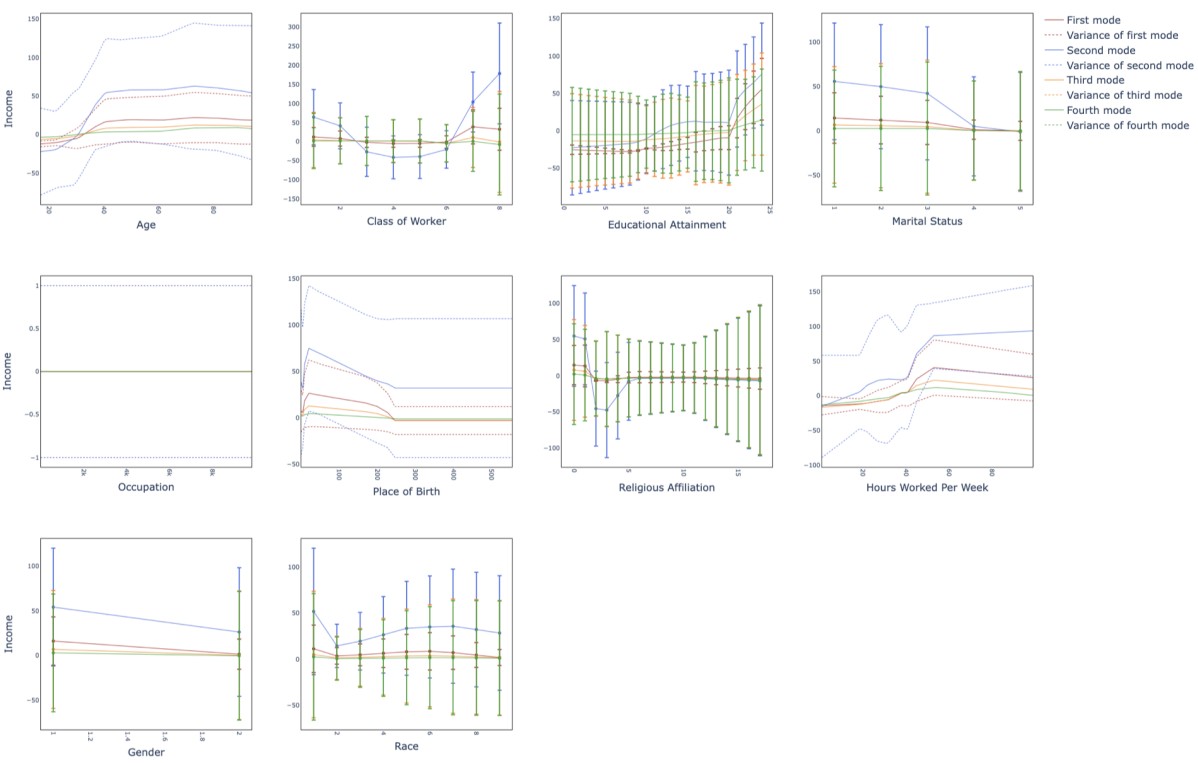

Figure 11: Learned relationships between features and labels for the MNAM on ACS Income datasets. Solid lines represent the mean of the relationships and dotted lines represent their uncertainties.

## C.5 ACS Travel

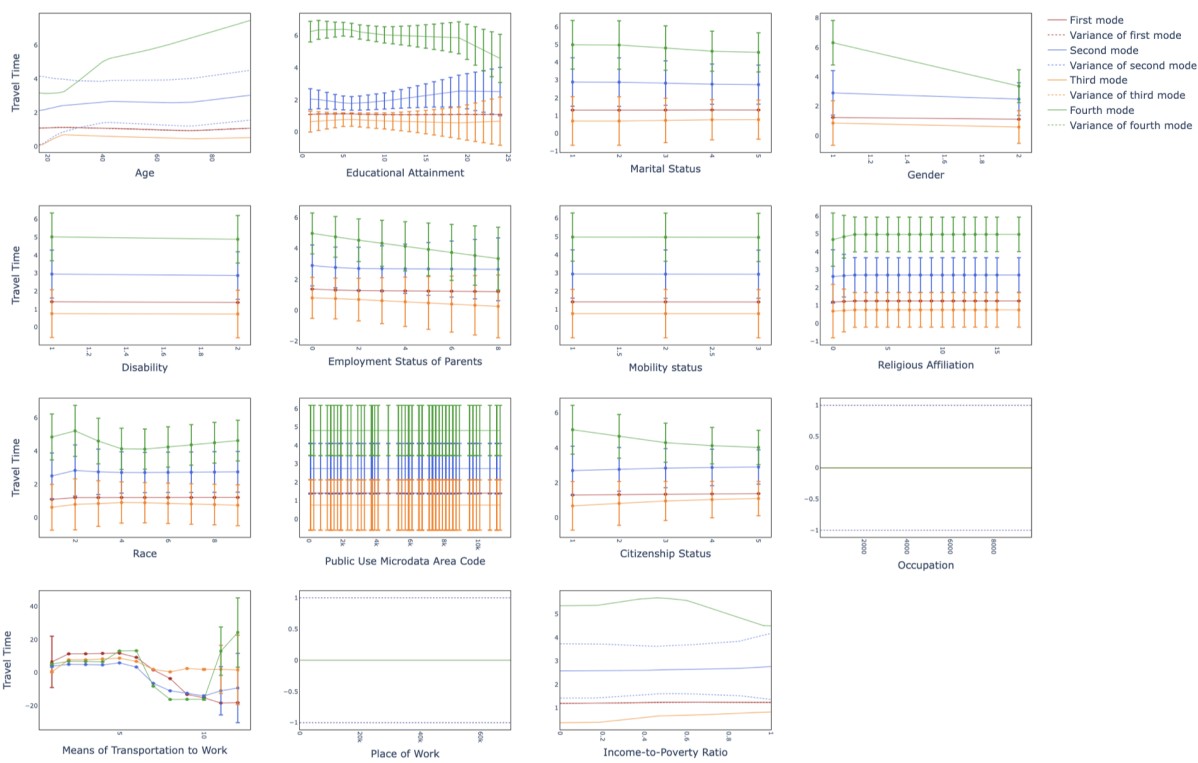

Figure 12: Learned relationships between features and labels for the MNAM on ACS Travel datasets. Solid lines represent the mean of the relationships and dotted lines represent their uncertainties.

# D   Relationship between feature and probabilities for a mixture of $k$ Gaussian distributions

In Figure 13, we examine the relationship between features and the probabilities associated with a mixture of $k$ Gaussian distributions. It is evident that the location variables have more significant impacts on altering probabilities compared to other variables, such as population and average number of beds. This analysis allows us to determine which features are pivotal in influencing the outcome of the neural network, which outputs probabilities for a mixture of $k$ Gaussian distributions.

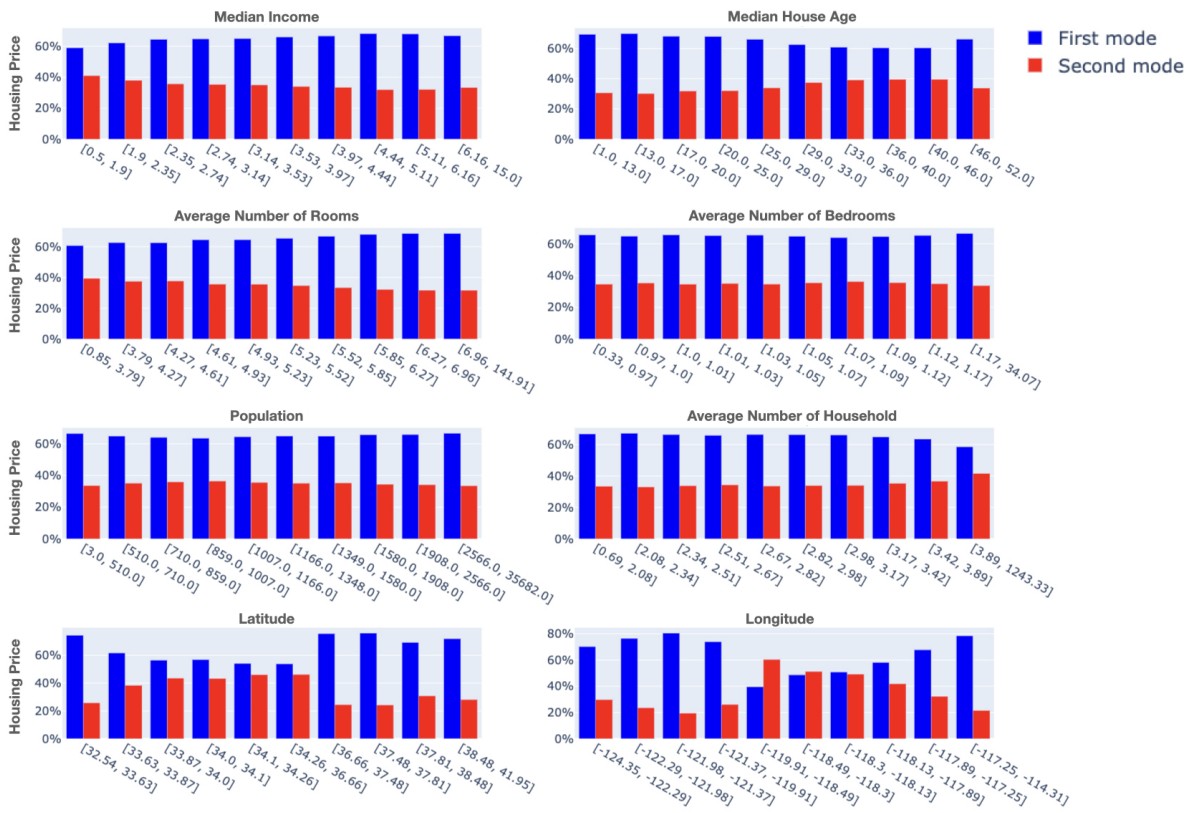

Figure 13: The relationship between features and probabilities in a mixture of $k$ Gaussian distributions. Each point on the x-axis represents a bin for the feature, which has been divided to ensure equal frequency. Each bar depicts the average probabilities for the mixture of $k$ Gaussian distributions across each bin.

