# OpenReview forum: "Generalizing Neural Additive Models via Statistical Multimodal Analysis"
_TMLR — Accepted by TMLR_

### Review · Reviewer_9cDv · 2023-10-30

**Summary Of Contributions:**

This paper proposes an extension to neural additive models (NAMs) to accommodate data that requires multi-modal, distributional predictions. The method involves predicting a mixture of Gaussians distribution, where each Gaussian is additive between features, and the mixture components are weighted according to a network that depends on all features. Ultimately, this is intended to provide a model (MNAM) that balances rich nonlinear predictions with the interpretability of NAMs.

**Audience:**

No

**Broader Impact Concerns:**

No broader impact concerns.

**Claims And Evidence:**

Yes

**Requested Changes:**

Several small issues:

- Typo in the abstract: GAN -> GAM.
- Citations throughout the paper lack parentheses where they should be used. The authors are perhaps using the \cite{} command rather than \citep{}.
- The authors use "multimodal" to refer to response variable distributions that are not well approximated by point predictions. It would be helpful to distinguish this from multimodality in the sense of using multiple training data types, e.g., text and images, perhaps in a footnote. This usage of term is popular today and could lead to confusion for some readers.
- Missing section link ("Section ??") on page 2.
- "log" should not be italicized in equation 2, and there should be more description leading up to this equation - it currently looks isolated.
- I agree that there could be advantages to the hard-thresholding training algorithm over the direct log-likelihood optimization, but I don't understand this sentence: "In contrast, MDN assumes the data is multimodal by nature but does not assume there is information in the input to learn the mode associated with a given sample." Training using the log-likelihood does provide a way to learn the mode associated with each prediction, because it would increase the weight on the mode that most accurately predicts the label. Unless I'm missing something, I think the authors' point here is incorrect.

**Strengths And Weaknesses:**

## Strengths

Compared to the previous version, there are several improvements:

- The comparison to MDNs and the discussion of training differences is helpful.
- The inclusion of a NAM with global variance parameter, and a pNAM with a single Gaussian mode are helpful. This makes it easier to see that 1) fitting a variance parameter and 2) fitting multiple modes are both helpful additions to the existing NAM.
- The inclusion of the log-likelihood metric, compared to the earlier usage of metrics designed for point predictions, is an improvement.

## Weaknesses

The premise of the paper is that NAMs and GAMs fail to represent multimodal relationships between predictors and response variables. This is somewhat confusing, because this problem exists for all mainstream regression models, whether they are interpretable or not: linear regression, random forests, gradient boosting and neural networks all output point predictions. So it's a bit confusing to focus on this issue in the special context of GAMs/NAMs.

Can the authors describe how they obtained confidence intervals for the results in Table 1? Ideally these would have been generated using multiple independent runs of each algorithm, but the confidence intervals are remarkably small in most cases, so that seems unlikely.

The utility of this extension to NAMs from a predictive point of view seems somewhat clear. In terms of the MAE and LL performance metrics, it seems like the MNAM provides a more flexible and therefore more accurate prediction. However, the utility from an interpretation perspective seems limited. The plots in Figure 3 are not simple to read, and this is for a case with only two modes and a couple features - it could become far more difficult with more modes. With this in mind, I have limited optimism about the use of mixture of Gaussian predictions and the HT training algorithm in this context, and I wonder if it would be more useful with generic DNNs rather than NAMs.

The proposed extension to classification problems using LIME still doesn't make sense to me. Unlike regression models, classification models automatically output distributional predictions because they assign a probability to each class. So it's unclear what a mixture would add here (a mixture of categorical variables is still a categorical), and the usage of LIME, a post-hoc explanation method, doesn't seem to make sense.

---

> ### Author Response · Authors · 2023-12-27
>
> We are thankful for the thorough revision and feedback provided by the reviewers. The paper is significantly improving thanks to the constructive comments.
> Next, we respond to each of their comments and highlight which changes have been included in the updated version of the manuscript (which are colored red in the updated pdf file for your convenience).
>
> ### Reviewer: The premise of the paper is that NAMs and GAMs fail to represent multimodal relationships between predictors and response variables. This is somewhat confusing, because this problem exists for all mainstream regression models, whether they are interpretable or not: linear regression, random forests, gradient boosting and neural networks all output point predictions. So it's a bit confusing to focus on this issue in the special context of GAMs/NAMs.
>
> The reviewer is correct in principle. We focus on NAM and GAM since they are among the most popular interpretable alternatives available, and in that context, we highlight their limitation. Our intention is not to claim these limitations are exclusive to these approaches, and as the reviewer correctly points out, are shared by other (less interpretable) alternatives.  We are more explicit and clearer with regards to this point in the updated version of the manuscript, see Section 4.
>
> ### Reviewer: Can the authors describe how they obtained confidence intervals for the results in Table 1? Ideally these would have been generated using multiple independent runs of each algorithm, but the confidence intervals are remarkably small in most cases, so that seems unlikely.
>
> We apologize for any confusion caused by not being more explicit on this. The remarkably small confidence intervals are a result of using an ensemble method. For our evaluation, we applied n-fold cross-validation, where 20 different models were trained. This was done by randomly dividing the training set into train and validation subsets for each fold. In the evaluation phase, we took the average predictions from these 20 models. Confidence intervals on each models were determined based on the mean and variance from the n-fold cross-validation, which has been clarified in Section 3.1.2.
>
> ### Reviewer: The utility of this extension to NAMs from a predictive point of view seems somewhat clear. In terms of the MAE and LL performance metrics, it seems like the MNAM provides a more flexible and therefore more accurate prediction. However, the utility from an interpretation perspective seems limited. The plots in Figure 3 are not simple to read, and this is for a case with only two modes and a couple features - it could become far more difficult with more modes. With this in mind, I have limited optimism about the use of mixture of Gaussian predictions and the HT training algorithm in this context, and I wonder if it would be more useful with generic DNNs rather than NAMs.
>
> In Figure 3, we have enhanced the readability by replacing dotted lines with highlighted areas to denote the uncertainty of relationships; this was necessary as an excess of lines could hinder users from effectively analyzing the plots. In addition, we offer plots in Appendix B that display only a single relationship per subplot to further enhance readability, though this approach does make comparing relationships more challenging.
>
> We agree with the reviewer in the sense that our proposed idea is more flexible (hence can fit better complex distributions), and that comes with a cost in interpretability (compared, e.g., to classic NAM as there are more relationships to interpret). We believe our method is of practical importance in practice, since is an “intermediate” alternative between very explainable (NAM) and very general (DNNs); in that sense, we believe our proposal fills an important gap and provides a viable alternative that is practical in complex applications that require multimodal modeling of data while providing a good degree of interpretability of the method. To acknowledge this aspect explicitly, we have updated the limitations discussion in Section 5.
>
> ### Reviewer: The proposed extension to classification problems using LIME still doesn't make sense to me. Unlike regression models, classification models automatically output distributional predictions because they assign a probability to each class. So it's unclear what a mixture would add here (a mixture of categorical variables is still a categorical), and the usage of LIME, a post-hoc explanation method, doesn't seem to make sense.
>
> We were inspired by LIME's ideas of locally approximating complex decision rules to provide (locally) explainable algorithms. We agree with the reviewer that the connection with LIME is still loose, and the proposed idea needs more work before it can be clearly outlined. For the reasons expressed above and in the interest of clarity we removed the potential connections and uses of LIME in the updated version of the manuscript.

---

> ### Author Response · Authors · 2023-12-27
>
> ### Reviewer: Requested Changes on those below small issues:
> * ### Typo in the abstract: GAN -> GAM.
> * ### Citations throughout the paper lack parentheses where they should be used. The authors are perhaps using the \cite{} command rather than \citep{}. +
> * ### The authors use "multimodal" to refer to response variable distributions that are not well approximated by point predictions. It would be helpful to distinguish this from multimodality in the sense of using multiple training data types, e.g., text and images, perhaps in a footnote. This usage of term is popular today and could lead to confusion for some readers.
> * ### Missing section link ("Section ??") on page 2.
> * ### "log" should not be italicized in equation 2, and there should be more description leading up to this equation - it currently looks isolated.
> * ### I agree that there could be advantages to the hard-thresholding training algorithm over the direct log-likelihood optimization, but I don't understand this sentence: "In contrast, MDN assumes the data is multimodal by nature but does not assume there is information in the input to learn the mode associated with a given sample." Training using the log-likelihood does provide a way to learn the mode associated with each prediction, because it would increase the weight on the mode that most accurately predicts the label. Unless I'm missing something, I think the authors' point here is incorrect.
>
> Thank you for those comments. We have addressed all of them in the revised papers.

---

### Review · Reviewer_qZmq · 2023-11-11

**Summary Of Contributions:**

The authors present a method that extends the Neural Additive Models (NAM) to handle multi-modal data distributions. Instead of producing a single output like NAM, the proposed model, Mixture of Neural Additive Models (MNAM), outputs the means, variances, and mixture membership probabilities of a mixture of k Gaussian distributions. The authors then presented two different ways to train the proposed model: the soft and hard thresholding algorithms. Finally, the authors demonstrate the benefits of the proposed model against several baselines, highlighting that the model provides a good tradeoff between accuracy and interpretability in datasets with multiple modalities.

**Audience:**

Yes

**Broader Impact Concerns:**

I do not have any broader impact concerns for this paper.

**Claims And Evidence:**

Yes

**Requested Changes:**

I would be grateful if the authors put some discussions about the question I mentioned above in the revision.

**Strengths And Weaknesses:**

Strengths:
- The idea of extending NAM to multi-modal area is very interesting.
- The author explained in detail how the model works and how the training procedure is conducted.
- Multiple experiments are conducted to explore different properties of the proposed model.
- The authors describe the connection of the proposed model to other related baselines and how the model is placed in terms of the accuracy vs interpretability tradeoff, as well as the limitation of the proposed model.

Weaknesses/Questions:

Overall, the proposed model is well explained in the paper. However, I have a few questions about the model.
- The model requires a parameter $k$ that describes the number of Gaussian distributions in the mixture. Are there any guidelines on how to choose it? Particularly in the case where we do not really know how much modality there is in our datasets.
- One of the downsides of a more complex model is that it may overfit the data. How does the model behave in the case where there are outliers? Does the model treat the outliers as another source of modality?

---

> ### Author Response · Authors · 2023-12-27
>
> We are thankful for the thorough revision and feedback provided by the reviewers. The paper is significantly improving thanks to the constructive comments. Next, we respond to each of their comments and highlight which changes have been included in the updated version of the manuscript (which are colored red in the updated pdf file for your convenience).
>
> ### Reviewer: The model requires a parameter that describes the number of Gaussian distributions in the mixture. Are there any guidelines on how to choose it? Particularly in the case where we do not really know how much modality there is in our datasets.
>
> This is indeed an important factor. Empirically, we observed that as long as the number of modes included in the model is larger or equal to the modes in the data, the network is able to train in an efficient fashion, and there is no noticeable harm in overestimating the number of modes (see for example results presented in Figure 6). Despite this, there is practical value from the perspective of interpretability to exactly match the number of modes present in the data. Currently, this is an issue we have not fully addressed. One option could be to start training models with many modes and decrease them until there is a significant gap in performance (similar to the “elbow method” used when determining the number of clusters). We added a comment regarding this issue in the updated version of the manuscript, Section 5, and explicitly acknowledged this limitation and space for improvement in the updated version of the Limitations Section.
>
> ### Reviewer: One of the downsides of a more complex model is that it may overfit the data. How does the model behave in the case where there are outliers? Does the model treat the outliers as another source of modality?
>
> Our model treats outliers as an additional mode, particularly when MNAM’s output consists of a large number of Gaussian distributions. In such cases, the model assigns one of the Gaussian distributions to the outliers. The effectiveness of this approach can vary by context. For example, in the medical field, recognizing a small but significant subpopulation that experiences severe side effects as a distinct modality could be critical for identifying key issues. However, this approach can be problematic if the outliers represent mere noise, underscoring the need for more objective methods of outlier identification. A key advantage of our model is that outliers do not impact Gaussian distributions to which they have not been assigned. Therefore, if outliers are accurately identified, they can be removed from the MNAM, mitigating any potential issues. This approach is elaborated upon in Section 5, where we added a more detailed discussion on this.

---

### Review · Reviewer_MoRB · 2023-12-14

**Summary Of Contributions:**

This works introduces MNAM, which extends NAM in two ways.  First, rather than having each of the per-feature neural networks predict a single value, MNAM has the each of those neural networks predict the mean and standard deviation of a gaussian distribution.  Second, rather than using a single gaussian for each feature, MNAM uses a mixture of gaussians. By learning when to use each gaussian component, MNAM can model feature interactions that NAM cannot (eg, if $X_1 = 0$, $X_2$ has a positive slope and, if $X_1=1$, $X_2$ has a negative slope).

**Audience:**

Yes

**Claims And Evidence:**

Yes

**Requested Changes:**

NAMs are generally considered interpretable because their structure makes simple for a user to 1) estimate the model's prediction for a new point and 2) estimate how the model's prediction would change if the point being explained was changed.  However, it's not as clear that this is possible for MNAM because of the $g$ function (ie, one can construct situations where all of MNAM's modeling power comes from $g$).  Please add discussion and/or experiments related to interpreting $g$.

A few potential ways to do this that came to mind are:
- How can the user tell if the subpopulations are differentiated by an observed or unobserved set of features?
- What do the plots in Figure 4/5 look like if you use a point drawn randomly from the dataset?  What if you first bin the dataset based on the feature on the x-axis and then show the weight of each component averaged across those bins?

**Strengths And Weaknesses:**

Strengths:
- The motivation behind adding the mixture model is compelling.
- For the tested datasets, MNAM fits the data better than NAM.

Weaknesses:
- The interpretability evaluation is based on a few worked examples.  Further, these examples don't really seem to discuss $g$ (the function that determines how to weight each of the gaussian components for a particular point) which seems essential to interpreting MNAM.

---

> ### Author Response · Authors · 2023-12-27
>
> We are thankful for the thorough revision and feedback provided by the reviewers. The paper is significantly improving thanks to the constructive comments. Next, we respond to each of their comments and highlight which changes have been included in the updated version of the manuscript (which are colored red in the updated pdf file for your convenience).
>
> ### Reviewer: NAMs are generally considered interpretable because their structure makes simple for a user to 1) estimate the model's prediction for a new point and 2) estimate how the model's prediction would change if the point being explained was changed. However, it's not as clear that this is possible for MNAM because of the g function (ie, one can construct situations where all of MNAM's modeling power comes from g). Please add discussion and/or experiments related to interpreting g.
>
> Similar to NAM, users can visualize how predictions change when the input components being explained vary as relationships are fixed. However, unlike NAM, the mode to which the sample corresponds may shift as other components change, due to the g function. As the reviewer points out this adds a layer of complexity compared to NAM. We acknowledge and describe this explicitly in the updated version of the manuscript in Section 5.
>
> ### Reviewer: How can the user tell if the subpopulations are differentiated by an observed or unobserved set of features?
>
> Since the proposed model has a probabilistic output (meaning, the likelihood of belonging to a mode is coded in the mode likelihood output), if the subpopulation can be recognized by an observed feature, the output for the given sample will “collapse” to one of the modes. For example, if we are modeling height vs. age and we recognize that two modes are present in the data (let us assume they are associated with the subject sex at birth), knowing the sex variable will collapse its sample output into its mode, if this input is not available, the model outputs modes probability that are set by the sex prior likelihood on the training data. We further discussed this in the updated version of the manuscript in Section 3.3.
>
> ### Reviewer: What do the plots in Figure 4/5 look like if you use a point drawn randomly from the dataset? What if you first bin the dataset based on the feature on the x-axis and then show the weight of each component averaged across those bins?
>
> In line with your recommendation, we have divided each feature into 10 bins of equal frequency. For the Housing dataset, we incorporated bar plots that display the weight of each component, averaged across those bins. Nonetheless, we acknowledge that the interpretability of the g function remains limited. We have discussed this limitation in the updated Section 5 of our paper.

---

### Decision · Action_Editor_qJo9 · 2024-01-20

**Recommendation:** Accept with minor revision

**Comment:**

Reviewers agree that the extension of the original NAM with mixture density network is interesting and useful in practice. Specific concerns about experiments are addressed by the author feedback, and the reviewers recommend acceptance for this submission.

In camera ready I suggest the authors to include all the provided edits in the current revised version. In the final recommendation, a reviewer further mentioned some related work -- pairwise GAMs (eg, https://www.microsoft.com/en-us/research/wp-content/uploads/2017/06/kdd13.pdf and subsequent work) and other interpretable ensemble models (eg, https://openreview.net/pdf?id=DdZoPUPm0a), for which the authors should consider adding at least a discussion regarding these relevant works.

**Audience:**

Deep learning community, explainable AI, statisticians interested in feature selection.

**Claims And Evidence:**

This paper presented a new model called Mixture of Neural Additive Models (MNAM), as an extension of the original NAM, to handle multi-modal data distributions. The main argument is to increase the expressiveness while still maintain interpretability for the NAM class of models.

Experiments clearly support the improved expressiveness claim. A few interpretability studies for the proposed model are provided.

---

> ### Author Response · Authors · 2024-02-01
> **Thank you**
>
> We extend our heartfelt thanks for the constructive feedback and insightful guidance provided by the action editor and reviewers during the review process. Your expertise and thoughtful suggestions have greatly contributed to the refinement of our manuscript.
>
> As advised by the action editor, we have added paragraphs on pairwise GAMs and interpretable ensemble models to our related works section. We have submitted the camera-ready version of our paper.
>
> Thank you again for everything.